# New Interpretations for Sprouting, Intussusception, Ansiform, and Coalescent Types of Angiogenesis

**DOI:** 10.3390/ijms25168575

**Published:** 2024-08-06

**Authors:** Alexander V. Korablev, Irina S. Sesorova, Vitaly V. Sesorov, Pavel S. Vavilov, Anna Mironov, Anna V. Zaitseva, Eugeny V. Bedyaev, Alexander A. Mironov

**Affiliations:** 1Department of Pathological Anatomy, Yaroslavl State Medical University, Yaroslavl 150000, Russia; 2Department of Anatomy and Topographic Anatomy, Ivanovo State Medical University, Ivanovo 153012, Russia; irina-s3@yandex.ru (I.S.S.); vit-sesorov@yandex.ru (V.V.S.); vavilov-007@mail.ru (P.S.V.); akb37@mail.ru (E.V.B.); 3Sacco Hospital, Universita degli Studi di Milano, 20122 Milan, Italy; anna.mironov@yandex.ru; 4Department of Anatomy, Saint Petersburg State Pediatric Medical University, Saint Petersburg 194100, Russia; zaytseva23@inbox.ru; 5Department of Cell Biology, IFOM ETS—The AIRC Institute of Molecular Oncology, Via Adamello, 16, 20139 Milan, Italy

**Keywords:** angiogenesis, endothelial cell, sprouting, intussusception, loop-based, inhibitors of angiogenesis, invadopodia, filopodia

## Abstract

Angiogenesis, or the development of blood vessels by growing from already-formed vessels, is observed in embryonic development, physiological cyclical processes such as wound healing, the encapsulation of foreign bodies, tumor growth, and some other situations. In this review, we analyze the cellular mechanisms of angiogenesis, namely, angiogenesis by sprouting, ansiform (by loop formation) angiogenesis, coalescent angiogenesis, and angiogenesis by intussusception (splitting the capillary into two channels). The analysis of data revealed a lot of unanswered questions and contradictions. Here, we propose several new models of angiogenesis explaining these contradictions.

## 1. Introduction

Angiogenesis is the process of the proper formation of a new vessel from a pre-existing one and the differentiation of the wall of a newly formed vessel. Angiogenesis—the formation of new blood vessels—is a hallmark of tissue repair, expansion, and remodeling in physiological processes, such as wound healing, ovulation, the restoration of the endometrium after menstruation, and embryo development, and in various pathologies, including cancer, atherosclerosis, and chronic inflammation. Knowledge of the mechanisms of angiogenesis is important for understanding how to treat angiogenic disorders and related diseases.

To date, most of the molecular mechanisms and machines involved in angiogenesis have been discovered. However, these molecular machines have been studied mainly in cell culture and are not tied to real structures in tissues. Thus, it is necessary to know not only the interactions of proteins but also the cellular and microanatomic aspects of angiogenesis and not only a set of proteins without their mapping [1]. In our opinion, the re-introduction of rather old results describing ansiform (by loop formation) angiogenesis is very important for scientists working in the field of angiogenesis. We think that the pioneering work in the field of angiogenesis should be known to our readers.

Here, we try to fill this gap, although we do not analyze physiological angiogenesis, angiogenesis in vitro, or angiogenesis under pathological conditions. Also, we do not examine the role of circulating endothelial progenitor cells because, among other reasons, such cells do not contribute to the regeneration of the arterial endothelium after an injury [2,3,4,5]. The progress in the analysis of molecular mechanisms involved in angiogenesis is outside of our focus because a lot of excellent reviews are available on the Internet. Moreover, we cannot produce our speculations about how these mechanisms operate in situ because these mechanisms were not tested under these conditions.

## 2. Sprouting Model

According to the current consensus [1,2,3], there are two types of angiogenesis: sprouting and splitting (intussusceptive angiogenesis). Usually, sprouts are formed on vessels lined with continuous endothelial cells (ECs), for instance, the parent capillary venule [6,7]. The first step in angiogenesis is the degradation of the basal membrane by matrix metalloproteinases, which disrupt the basement membrane (BM) [8]. The cells composing the endothelial sprout have also been described as having activated cellular organelles and a fragmented BM [9,10,11,12].

Usually, these leading ECs are formed after mitotic division [13]. However, Sholley et al. [14] reported that a limited number of endothelial sprouts were observed even when endothelial proliferation was suppressed by irradiation. ECs synthesizing DNA are visible in the walls of both the forming sprouts and maternal vessels. There are no dividing endotheliocytes at the top of the sprouts [1]. Whether stimulators of angiogenesis induce endothelial mitosis directly or whether they act solely as chemotactic stimuli of the endothelium is as yet unknown [11,15,16,17]. Before the appearance of capillary sprouts, mitotic figures considerably increase in the vascular endothelium. Stalk cells do not extend protrusions [18,19]. Burger et al. noticed increased endothelial DNA synthesis beginning before capillary sprouts [15].

Rhodin and Fujita [20] described capillary sprouts originating from arteriolar–venular arcades. A solid sprout tip was progressively lengthened in alternating rapid and slow growth phases. The formation of the sprout occurs through several stages. The process starts when the local concentration of a ligand or several ligands for EC receptors located on the basolateral plasma membrane (BLPM) of ECs of the maternal vessel is increased. This local site could be a tumor or a focus of a chemical, for example, AgNO_3_ [21], or other types of tissue damage, where angiogenesis will then begin. It is not clear why angiogenesis occurs under the action of silver nitrate. It seems that the destruction of cells, including silver, leads to the release of lysosomal enzymes into the interstitial space. Lysosomal enzymes begin to cleave matrix proteins into fragments, some of which turn out to be similar to the active domains of growth factor proteins.

Then, the diffusion of a ligand or several ligands toward the microvessel begins from this focus (usually a postcapillary venule or capillary lined with a continuous endothelium). Growth factors secreted by damaged cells, or tumor cells, are usually described as ligands. Some growth factors predominantly stimulate the angiogenesis of blood vessels, while others cause lymphatic vessels to grow.

The EC-receptor ligands arriving at the BLPM bind to their specific receptors present on the BLPM and induce the formation of dimers from identical receptors. Dimerization, in turn, leads to the activation of the kinase activity of cytosolic receptor domains, which begin to phosphorylate proteins associated with actin polymerization. The polymerization of actin group proteins leads to the formation of fibrils, which begin to put pressure on the BLPM, and the BLPM protrusion is formed.

The sprouts grow toward each other from nearby vessels on opposite sides and merge. Then, the APM appears, and the lumen and blood flow are restored. In each sprout, a single tip cell pioneers vessel path finding; this cell is highly mobile and tubeless and proliferates minimally or not at all. Endothelial stalk cells follow the tip cell, proliferate to form an elongating stalk, and create a lumen [18]. Such changes stimulate the ECs of the maternal vessel to divide. ECs enter the S and then the G2 phase of the cell cycle and finally divide [22].

Further, the consensus is that the protrusion or sprouts are channeled, especially when sprouts growing toward each other meet and contact each other. It is not clear from the consensus information whether new APM formation or the restoration of the vessel lumen is necessary. It is not clear whether the leading EC breaks away from the reservoir with a complete loss of the apical PM (APM) and tight junctions. Alternatively, the APM and tight junctions may persist. If there is only one leading EC, then it remains unclear why the reciprocating movement of red blood cells was observed by Rhodin and Fujita [20] in the growing sprouts. Also, there are no BMs around new capillaries [1]. The BM is synthesized during the G2 phase [22,23].

In tumors, vessels are lined by a disorganized, discontinuous, pseudostratified EC layer, with ECs crawling over each other while forming gaps or being absent in other denuded areas; ECs protrude filopodia-like extensions in the lumen (these are not filopodia, because their diameter is equal to 1.07 µm), possess numerous fenestrations (the diameter of these openings is equal to 670 nm), and fail to form tight cellular junctions but are separated by intercellular gaps [24,25]. The ECs from which the protrusion of the BLPM is formed become the tip ECs [20].

Next, the reorganization of the new capillary begins: newly formed due to the division of old pericytes, new pericytes move there and also secrete a BM. The mechanism of the extension of EC nuclei that form the sprout into the wall of the sprout is not clear. It is not clear what causes the division of ECs in the wall of the maternal vessel after the appearance of a sprout. Also, it is still not known when the BM is formed and how the intracellular transport of collagen IV and other proteins necessary for the formation of the BM is organized in ECs [24,25,26,27]. Figure 1 and Figure 2 show possible mechanisms of sprouting. In Figure 2, we depict only a very rough scheme related to cell signaling during sprouting.

## 3. Stimulators and Inhibitors of Angiogenesis

Many reviews describe stimulators of angiogenesis [25,26,27]. Here, we have neither the space nor the wish to repeat these well-known facts. For instance, equine aortic endothelial cells show pro-angiogenic behaviors in response to FGF2 but not VEGF-A [25]. VEGF-A controls angiogenic sprouting in the early postnatal retina by guiding filopodial extension from specialized endothelial cells situated at the tips of the vascular sprouts. The tip cells respond to VEGF-A only by guided migration; the proliferative response to VEGF-A occurs in the sprout stalks. These two cellular responses are both mediated by the agonistic activity of VEGF-A on VEGF receptor 2. Whereas tip cell migration depends on a gradient of VEGF-A, proliferation is regulated by its concentration [26,27,28].

On the other hand, more than 60 angiogenesis inhibitors (AIs) have been tested. Some of them are isolated from cartilage. Monoclonal antibodies against EC growth factors could be useful. Under certain conditions, the properties of AIs have tumor necrotizing factors-alpha and -beta, thrombospondin, platelet factor IV, transforming growth factor-beta, and interferons. Cortisone, dexamethasone, and adrenocorticotropin have properties of AIs. Medroxyprogesterone, a synthetic steroid that blocks collagen synthesis, has similar properties. The properties of AIs make them specific inhibitors of collagen deposition (proline analogs) and inhibitors of hydroxyproline synthesis. The inhibition of collagen cross-linking by β-Aminopropionitrile inhibits angiogenesis. The regression of growing microvessels with the help of proline analogs and prolyl hydroxylase inhibitors is realized by blocking the formation of the three-dimensional structure of collagen fibrils.

The mechanism of action of angiostatic steroids is associated with the fact that they inhibit the accumulation of collagen, causing the destruction of the BM. Also, actin and tubulin inhibitors are angiogenesis blockers [26,27,28]. Targeting tumor vessels by blocking VEGF has become an established anticancer strategy [18]. Angiogenesis is modulated by syntaxin 6, a Golgi- and endosome-localized t-SNARE. Syntaxin 6 and alpha-5 beta 1-integrin are localized in EEA1 containing early endosomes, and the functional inhibition of syntaxin 6 leads to the incorrect direction of alpha-5 beta 1-integrin into lysosomes and a decrease in cell proliferation via fibronectin [29].

## 4. Intussusceptive Angiogenesis

Another mechanism is intussusceptive angiogenesis, which represents a comparatively new mechanism described by Burri’s group [30,31]. According to this model, pre-existing vessels split or are remodeled through the formation of transluminal tissue pillars, leading to the expansion (intussusceptive microvascular growth), arborization (intussusceptive arborization), or branching remodeling (intussusceptive branching remodeling) of the pre-existing vasculature. This type of angiogenesis has been described in different animal models during both normal and pathological microvascular growth; during the development of the lung, kidney, ovary, retina, bone, and skeletal muscle, among other tissues and organs, e.g., in the rapidly expanding pulmonary capillary bed of neonatal rats; in different organs; and even in tumors. This process starts when contact between the APMs of ECs from opposing capillary walls is formed. Then, inter-endothelial cell junctions are subjected to reorganization, and an interstitial pillar core is formed that is invaded by pericytes.

By this stage, transluminal pillars have a diameter of about 2.5 µm. Then, a hole through this contact is formed. Continuous pillar formation and growth lead to a rapid expansion of the capillary plexus. Additionally, intussusceptive microvascular growth permits the rapid expansion of the capillary plexus, furnishing a large endothelial surface for metabolic exchange. Intussusceptive arborization causes changes in the size, position, and form of preferentially perfused capillary segments, creating a hierarchical tree [32,33,34,35,36,37,38,39,40].

When a blood capillary is confined to a single EC, the EC is referred to as a seamless EC. The number of SEs is roughly correlated with the number of branches in the capillary system. Seamless ECs make up about 50% of all capillaries in the renal glomeruli. Seamless ECs are found in arterio-venous capillaries in the capillaries of endocrine glands, as well as in the sinusoidal systems of the heart muscle, liver, spleen, and bone marrow. As glomerular maturation proceeds, the capillary loop becomes divided into six to eight loops. Most of the capillaries in the glomeruli of the kidney are seamless. This suggests that during the differentiation of these glomeruli, the main mechanism of angiogenesis is splitting [1,41,42]. Terasaki et al. [43] demonstrated that during glomerulus maturation, both intussusceptive angiogenesis and sprouting are used. Also, it was shown that not only intussusception but also coalescent angiogenesis is an important mechanism of tumor angiogenesis [44].

Several molecules and blood flow participate in the regulation of intussusceptive angiogenesis. The variable expression of VEGF and VEGF receptors, namely, VEGFR1 and VEGFR2, may regulate sprouting or intussusceptive angiogenesis [45,46]. Hypoxia-inducible factor 2 alpha, angiopoietin 1 and 2, FGF (fibroblast growth factor 2), and platelet-derived growth factor beta (PDGF beta) also act in intussusceptive angiogenesis regulation. Blood flow changes play an important role in vascular plexus remodeling and the differentiation of venules and arterioles [47,48,49].

The development of intraluminal pillars is an important step in intussusceptive angiogenesis (Figure 3A and Figure 4A,B). The exact mechanism of pillar formation is unknown. Intraluminal nascent pillars that contain a collagen bundle covered by ECs in the vasculature of experimental tumors have been observed [35]. Paku et al. [35] proposed a new mechanism for the development of these structures and called this process inverse sprouting. Initially, intraluminal endothelial bridges are formed. Next, the BMs of the ECs in the area of the bridge are dissolved. Then, a pulling force is exerted by the actin of the ECs via specific points, which attach to the collagen bundle and contain vinculin. Finally, the collagen bundle is moved into the pillar, and the deposition of new collagenous connective tissue occurs (Figure 4B). Recently, a new mechanism of intussusceptive angiogenesis was proposed. This process is driven by the formation of a core initially formed by a microthrombus, which was replaced by an extracellular matrix and invaginating pericytes, establishing numerous peg-and-socket junctions with ECs [50]. Finally, we propose a new model of this type of angiogenesis (Figure 3A and Figure 4A). It is based on our analysis of this type of angiogenesis in intestinal villi (see Zaitseva et al. in the current issue).

## 5. Ansiform (by Loop Formation) Angiogenesis

The fundamental role of ansiform growth in the development (angiogenesis) of the intra-organ circulatory system was shown a long time ago [52,53]. Briefly, in the early stages of prenatal ontogenesis, the large omentum is dominated not by primary capillary networks but by vascular pairs consisting of arterial and venous trunks that absolutely accurately copy not only the course but also the branching order of each other (Figure 3B–K and Figure 4C–I). There are ascending and descending arms of the loop. We believe that hemodynamics does not stop in loops.

Each such pair ends in a loop (the place where blood passes from the bringing vessel to the taking one) with a growth bud at the top. The tip of such a loop has an outer arc and an inner one. The growth of the vascular loop occurs due to the coordinated work of both arches, which are carriers of the external and internal growth buds. The outer growth bud ensures the development of the developing tissue of the omentum, and the inner one performs a dividing function, providing elongation of the arterial and venous knees of the loop.

The loop-shaped vessel, as it grows, is able to form a daughter-like vessel. The zone of formation of daughter loops is the tip of the maternal loop, where the outer and inner growth buds are located on the same axis. The maternal loop-shaped vessel responds to the influence of growth inducers not by changing the direction of growth but by forming a new sprout on the outer arc, which forms a domed bulge passively filled with blood. The external maternal arch continues its growth. Sprouts became divided it into two sections: one is connected to the arterial knee, and the other to the venous knee of the maternal loop. As a result, while maintaining the continuity of blood flow, the rudiment of a daughter loop is born, which develops in the direction of the growth inducer. If the daughter sprouts arise not along the middle line but asymmetrically, then a single daughter vessel is formed and, as a rule, from the arterial part of the loop. Loop-forming angiogenesis is characterized by the following properties: the absence of a temporary gap between the growth of blood vessels and blood flow and the possibility of the proliferation of endothelial cells of vascular loops with continuous blood flow (Figure 3B–K).

The outer growth bud of one loop can connect to the growth bud of another loop, and a cytoplasmic bridge arises between them. Gradually, this connection becomes channeled under the influence of blood flow in both loops, which means the birth of the first vessel of the capillary structure, which is called the connecting capillary. The direction of blood flow in it depends on pressure fluctuations in the vascular pairs that it combines. A single daughter vessel is most often formed from the venous knee of the loop. The number of connective and main capillaries in the circulatory network of the organ gradually increases. The appearance of connections between the vessels and main capillaries marks the completion of loop-like growth for these specific vascular loops, since the tip of each of them loses the ability to form daughter loops (Figure 3B–K, Figure 4B–H and Figure 5).

The development of the circulation bed in human embryogenesis is brought about by a loop-like growth of vessels manifesting in the proliferation, integration, and morpho-functional transformation of the vascular loops. The proliferation of the loops ensures blood inflow and reflux at all levels, while their integration is followed by the formation of the hemo-circulation system [52,53].

Patan et al. [54] revealed that a new vascular network composed of venous–venous loops of varying sizes grows inside a tumor from the wall of the adjacent main vein. Intussusceptive microvascular growth is responsible for this. However, this mechanism is rather different from the ansiform growth of blood vessels. In his subsequent paper, Patan [55] states that angiogenesis comprises two different mechanisms—endothelial sprouting and intussusceptive microvascular growth—but does not mention the formation of loops by newly formed venous vessels.

Additionally, it is necessary to mention a special type of angiogenesis that occurs in skeletal muscles with their constant contraction during electrical stimulation or during the training of athletes or prolonged exercise by running animals [56,57,58]. Interestingly, in most modern papers, cellular events related to exercise-dependent angiogenesis are not described. Classic works by Hudlicka et al. [57] showed that after prolonged exertion, muscle fibers thicken, and the number of capillaries in muscles increases dramatically. At the same time, the vessels of the arterial and venous links expand. After a certain period of adaptation, the process reaches a plateau. At the same time, during the adaptation process, signs of capillary cleavage and profiles are found in the microcirculatory bed, which the authors describe as sprouts. In the projections of the capillaries, round holes and even narrow slits are visible, characteristic of cleavage angiogenesis, as shown in figure 1C,D presented by Hansen-Smith et al. [58]. Judging from the presented images, in addition to the usual capillaries, capillaries with a sharply reduced number of caveolae appear. These features of ECs are typical of EC sprouts.

In addition, there are slit-like lumens. In papers describing angiogenesis under such conditions, there are no images showing sprouts growing toward each other. Vessels with a narrow slit-like lumen connect parallel longitudinal capillaries. Thickened myocytes compress and flatten blood capillaries, triggering the mechanism of splitting angiogenesis. Therefore, along with sprouting, there is a process of vascular splitting. Loops typical of splitting angiogenesis are visible in figure 5b presented by Hudlicka [57]. Signs of capillary splitting and loop-like sprouts are visible in muscles subjected to exercise-dependent stress. For instance, in figure 6 presented by Hansen-Smith et al. [58], there are features of intussusception similar to those proposed in the paper by Zaitseva et al. (2024; current issue).

## 6. Filopodia or Invadopodia

One of the main problems in sprouting is understanding the nature of these long processes protruding from the tip ECs and nearest ECs. According to the consensus, these protrusions are called filopodia. Carmeliet et al. [18] state that tip ECs at the forefront of vessel branches are highly polarized cells that use filopodia to guide a sprouting vessel toward an angiogenic stimulus [18].

However, usually, the filopodium is understood as a cylindrical outgrowth of a plasmalemma with a diameter of 100–120 nm. Their diameter is uniform. Filopodia are filled with bundles of 7.5 nm actin filaments. Filopodia do not contain microtubules. Our measurements of the outgrowths for each type of EC presented in the original articles show that their diameter varies from 600 nm (less often 300 nm) to several micrometers. For example, the diameter of the protrusion shown in the EM images is 3–4 microns. Finally, these outgrowths must be able to dissolve the BM, that is, have metalloproteases at their tips or secrete secreted proteases in some way. However, such a property of these outgrowths is not described in the literature. It is more likely that these outgrowths of the PM represent invadopodia [59].

The sprouts should form contacts with the lateral walls of intact vessels. How this happens is not clear. However, we assume that the sprouts have the properties of invadopodia, which can destroy the BM and form contacts with the BLPM of ECs of the longitudinal vessel. In addition, after the adaptation, there are many loops in the vascular bed. It seems to us that, in this case, too, the sprouting model—based on the assumption that sprouts are formed by two ECs and that, as a result of the sewerage or canalization of such a sprout, a vascular loop will form from which another such sprout will grow—is quite possible. Invadopodia consist of 200 nm wide and up to 3 µm long membrane protrusions that extend into degraded areas of the extracellular matrix. This protrusion could be inserted into the cytoplasm. In this case, it is surrounded by the cap-like invagination of the PM [59]. Therefore, during angiogenesis, ECs form not filopodia but, most likely, invadopodia. Invadopodia have the ability to secrete metalloproteases and mechanically pierce the BM and other structures of the extracellular matrix. Also, Rhodin and Fujita [20] described capillary sprouts. The cytoplasm of the protrusion contained an array of microtubules, 7.5 nm filaments, and many small vesicles. Thus, protrusions evolving from the leading PM of EC leaders are not filopodia but invadopodia.

Thus, when angiogenesis of blood microvessels takes place in the cornea, there are no filopodia: there are invadopodia and many vascular loops. When angiogenesis of blood vessels is studied in the retina, there are many filopodia. There are loops, too. If this process is studied in a three-dimensional gel in vitro, then filopodia are visible, but there are no loops. Indeed, in an in vitro model of angiogenesis, ECs do not form loops (images presented in [1]).

It seems that during angiogenesis in the retina, filopodia exist because, in the brain and retina, there are no extracellular matrix proteins or filaments and there is no necessity to dissolve matrix proteins, as occurs in vitro when three-dimensional gel is not very dense. Movies presented by Kamei et al. [60] show that during prenatal angiogenesis, EC processes can emanate from an EC when the EC already has the apical domain. These thin protrusions are formed from the BLPM of the tip but also from the stalk ECs. In movies presented by Kamei et al. [60], processes that quickly grow and then undergo retraction have been observed. Light microscopy cannot tell us about the thickness of these protrusions because, for two-photon microscopy, the resolution is restricted to 300 nm. Unfortunately, the three-dimensional EM reconstruction of these protrusions was not presented. However, in the cornea, filopodia cannot be formed due to the significant resistance of the interstitial matrix. Thus, protrusions cannot be filopodia. After the dissolution of the BM, these processes penetrate through the BM and induce the movement of ECs through the interstitial space.

The next important issue is the issue of the structure of the outgrowth or protrusion on the BLPM of the EC. Images of those rare cases where sprouts were studied using an electron microscope showed that, in most cases, especially on serial sections of 3VIEW, these outgrowths consist of two ECs, between which there is a lumen similar to the APM. Moreover, this single slit of the lumen is usually divided into two segments by an island in which two ECs are tightly attached to each other, and adhesive compounds are found here. Two similar gaps are visible in the figures presented by Ausprunk and Folkmann [11].

These observations suggest that, in vivo, ECs do not lose their PMs and do not depolarize. Moreover, most likely, the appearance of a new lumen is not required, since it always exists. And, if so, then the islet of adhesive compounds of the APM of two ECs in the sprout can become a way to allow blood circulation to begin around this islet without the need to form a new lumen (see the scheme in our Figure 1).

Filopodia protruding from tip ECs are visible only in vitro in a 3D matrix and in the retina. However, these types of cell processes are not found in the cornea. Vascular loops are observed in the cornea and retina, but these loops are observed rather rarely in vitro. Filopodia have an apical membrane, and the formation of filopodia leads to invagination due to TAA. There are no filopodia, but there are narrow lamellipodia or IVP. It is necessary to dissolve the BM, that is, by metalloproteases. The outgrowth on the BLPM, apparently the invadopodium, is visible in figure 10 presented by Ausprunk and Folkman [11]. Also, figure 7 presented by Ausprunk and Folkman [11] shows an invadopodium, whereas figures 8 and 10 presented in the same paper demonstrate two lumens.

In tissue culture of the aorta, the treatment of tissue samples with TGF-beta induces the formation of functionally active invadopodia by the BLPM of ECs [61,62]. The images in the paper by Zhou et al. [63] suggest that the penetration of endothelial protrusions through the BM is based on the generation of IVP.

In the retina, there is no extracellular matrix, only neurons and glial cells. Figure 2k presented by Gerhardt et al. [64] shows the loop-like bloodstream with the island in the center. It is not clear whether real filopodia are formed. However, in the nervous system, the extracellular matrix is absent between neurons and glial cells, and the penetration of the cellular protrusion is much easier than in other tissues. Tip cells were not unique to the retina but were present in other parts of the developing mouse central nervous system harboring a similar pattern of active angiogenesis (unpublished data by Gerhardt et al. [64]). In spite of this, Gerhardt et al. [64] call protrusions emanating from tip ECs “filopodia”. They write that endothelial filopodia emerging from the tip ECs in the postnatal retina were uniform in thickness (100 nm) but of variable length, with the longest extending >100 µm. However, they used light microscopy resolution, which is restricted to 200 nm. Moreover, in their images, the width of so-called filopodia is composed of more than 1 pixel. Finally, they did not check whether these structures contain tubulin and internal membranes. It seems that these protrusions represent processes emanating from the BLPM.

Additionally, tip cells with protrusions already have the apical domain integrated into the luminal surface of the capillary (see figure 2h presented by Gerhardt et al. [64]), suggesting that so-called filopodia represent protrusions from the BLPM. These processes are rather thick and probably contain microtubule and membrane structures, as has been shown by Rhodin and Fujita [20], that are transported to their tip for growth. As such, the similarity of these protrusions to growing neurites is only phenomenological because neurites represent the apical domain of neurons and thus should have different mechanisms of membrane delivery.

## 7. Movement of Blood

Rhodin and Fujita [20] observed the oscillatory movement of red blood cells in the capillary sprout. However, if sprouts represent blind tubes, then oscillatory movement is impossible without the extravasation of plasma. Wakui et al. [65] demonstrated that an EC sprout of about 8 µm in length is composed of two ECs. Its lumen had already made contact with the parent capillary lumen through three-dimensional aspects. The endothelial sprout lumen is an elongation and/or extension of the parent capillary lumen. It was revealed that the endothelial sprout already possessed cellular polarity and luminal and abluminal cell surfaces. This suggests that the endothelial sprout grows in a bicellular configuration, being composed of a pair of relatively immature ECs, which extend and migrate outward, in keeping with an intercellular slit-like lumen to connect the parent capillary lumen. ECs proliferate, producing a pair of daughter cells. While these daughter cells are in the wall of the parent capillary, a new lumen evolves. The new lumen connects to a portion of the parent capillary lumen in the very early phase of the formation of the sprout. The endothelial sprout is further distinguished by the migration and/or extension of the endothelial cytoplasm in a bicellular configuration. Throughout the whole process of endothelial sprout formation, pericyte–endothelium interaction, conducted by the two cells cytoplasmic caving in on each other, seems to have an important role in sprout growth, probably through a regulatory control mechanism in which they act on each other. The configuration of these cells was nearly symmetrical two- and three-dimensionally and was well demarcated from neighboring endothelial cells by its contour, contents of cellular organelles, and immaturity [66] (Figure 5).

## 8. A Model Based on the Use of the Tips of Vessel Loops

Carmeliet et al. [18] show schemes where two microvessels are situated opposite each other. However, in most cases, sprouting takes place from only one side, especially in the cornea and retina. Indeed, in most assays used for the study of angiogenesis (damage to the cornea due to silver nitrate or its cauterization, the introduction of a gel containing growth factors from the thickness of the cornea, the study of angiogenesis in the growing retina…), there is a unilateral growth of sprouts toward the source of angiogenesis stimulants, which, in most cases, are ligands of growth factor receptors located on the BLPMs of ECs. There are no sprouts growing toward each other, although they are found everywhere in angiogenesis diagrams shown in literature reviews that are devoted to angiogenesis.

The formation of sprouts occurs at the site of inflection of the vascular loop. Our earlier views do not answer the question of how a new loop can form on a smooth section of the vessel, where blood flow in the arterial and venous knees of the capillary loop goes in opposite directions. On the one hand, this is possible due to the fusion of the lateral surfaces of two sprouts simultaneously growing in the same place in the arterial and venous knees of the capillary, or this loop forms on the venous knee (see below). But, it is not clear why such synchronicity occurs. It can be assumed that the sensitivity to ligands in the BLPM EC of the arterial knee is lower than in the EC of the venous knee.

The initial events during sprouting and vascular-loop-dependent angiogenesis are described rather purely. Only some papers are based on electron microscopy [11,13,20,65]. In the most important paper by Wakui [65], the thickness of the process composed of two ECs is 3.6 to 4.8 microns. In this paper, figures 5, 8, 9 and 10 presented by Wakui [65] demonstrate that gaps between the two ECs are already visible, separated by the attachment island, which represents the close apposition of two apical plasma membranes of opposing ECs. Interestingly, figure 3 presented by Ausprunk and Folkman [11] shows a vascular loop.

We believe that the formation of EC processes, which have great similarity to IVP, occurs in the area of intercellular contacts of ECs on the convex surface of the loop, especially where the contact is extended along the blood (Figure 5). The BM is less pronounced here, and the motor activity of actin and the edges of ECs are higher. During the formation of a sprout, the IVP is pulled out simultaneously by two ECs, which have a slit-like cavity between them. The middle part of this gap turns out to be very narrow. Here, due to adhesive joints, the ECs are tightly attached to each other.

They stick to each other. Around this zone, the inter-endothelial gap is slightly thicker and does not contain adhesive compounds. From the periphery, the entire gap is limited by gradually forming dense junctions. For some time, the gap and the lumen of the vessel are separated by a thin bridge of the cytoplasm of the EC, with dense connections between the ends of the EC. This bridge gradually becomes thinner and breaks, and the gap is immediately filled with blood, which can move in the forward and reverse directions around the island with adhesive joints. This movement was discovered by Rodin and Fujita [20]. For some time, in the bend of the maternal loop, there are, as it were, two channels through which blood moves, but the new channel expands, and the opposite (from the concave side) wall of the maternal vascular loop gradually approaches the islet and merges with it. And, again, there is only one channel left. We believe that our hypothesis allows us to reconcile two observations, ours [51,52,53] and that by Rodin and Fujita [20], who, during a lifetime study of growing mesentery microvessels, recorded the oscillating movement of red blood cells in the sprout. We cannot exclude the absolutely simultaneous occurrence and then fusion of sprouts on both (both on the ascending and descending trunks of the loop). It is quite possible. But, from the current position, the first model seems to us to better explain the available EM images.

Tight junctions should form between the two ECs in the sprout. They are formed only in the presence of two contacting ECs. If they are not present, then there will be a blood leak into the interstitial space as it passes through the sewerage.

We propose the following hypotheses (Figure 1): 1. During angiogenesis and sprouting, the ECs undergo complete or partial depolarization with the intermixing of the APM and BLPM or the elimination of the APM from the external surrounding EC membrane. 2. The sprouts can grow only from the BLPM or from the already-intermixed PM. 3. After sprouting, ECs should undergo differentiation with a short stage of procollagen synthesis and transport for the generation of the BM. It should be procollagen-containing distensions during this stage. 4. During angiogenesis, the intracellular vacuole composed of the APM is formed not from caveolae but from previously formed vacuoles where the APM is extracted during depolarization (Figure 3). The outgrowth on the convex part of the inflection consists of two ECs. If there are few stimuli for angiogenesis, then loop-like growth will occur, and if there are many such stimuli, then sprouting will be prevalent. Outgrowths are formed from near-contact zones. If we assume that any sprout is formed by the protrusions of two ECs, it is easier to channel and explain the reciprocating movement of red blood cells. If two ECs on one side and two ECs on the other side are in contact, then a lumen is quickly formed.

Unclear issues of intussusceptive angiogenesis can be resolved if we recall that this type of angiogenesis is found in organs where an increase in capillary density is required in a limited volume. And, in these cases, there is no need for protrusion formation: the lumen is just squeezed, and the APMs are glued to each other. Then, there are all those processes described above. In this case, blood pressure in the lumen of the microvessel can lead to the expansion of the APMs of two cells, which leads to the formation of a protrusion.

Finally, the idea that the sprout consists of two ECs separated in the middle by an island of PM attachment containing adhesive junctions allows us to answer the question of loop-like growth and assume that a new loop forms on the venous part of the capillary loop, as described above, and it is this loop that is found when studying the omentum (Figure 5). In order to explain the formation of loops in tissues with the unidirectional growth of the angiogenic mass toward the source of growth factors, a mechanism of fusion of sprouts by lateral surfaces is required. When using the plastic cast method, holes in the tops of the loops are visible in the cornea on the casts. Where the vessel dilates and where the blood flow turns 90–180 degrees, the pressure on the ECs increases, and they respond better to the action of angiogenetic ligands. In the images (figure 1i) presented in the article by Jakobsson et al. [66], holes in the microvascular bed similar to those observed during intussusception are observed. Also, it is clearly visible that two ECs form the tip of the sprout. However, in this paper, EM analysis was not performed, and we propose that if it had been, these data would be similar to those obtained in the paper where two ECs were visible on the tip of sprouts. As such, Jakobsson et al. [66] discovered not a competition between tip ECs but actually an observation indicating that every sprout is composed of two ECs.

Although it is assumed that signaling mechanisms are well known, in reality, these mechanisms were obtained in vitro and often even using other cells (not ECs). These mechanisms were not tested in organisms. For instance, in a very recent review on angiogenesis during wound healing [67], signaling mechanisms are not even mentioned because it is a complex process that involves the coordinated actions of many different tissues and cell lineages and requires the tight orchestration of cell migration, proliferation, and matrix deposition and remodeling, alongside inflammation and angiogenesis. The authors stressed only that the angiogenic response commences within hours of injury, with sprouting vessel tips eventually forming a capillary bed many times denser than that of adjacent normal tissue. It was shown in zebrafish that severed vessels elongate and anastomose with each other and later become tortuous through the proliferation of endothelial cells [68]. Damaged epithelial cells and macrophages release angiogenic ligands. The dominant pro-angiogenic factor during wound healing is VEGFA. Other less important factors are FGF2 and pigment epithelium-derived factor. In mice, the deletion of macrophages during the early stages of the wound response led to the reduced formation of vascularized granulation tissue [67,69,70].

## 9. Conclusions

The initial events during sprouting and vascular-loop-dependent angiogenesis are described rather purely. Here, we present hypotheses explaining evidence in favor of the loop-like canalization of sprouts. In our paper, there are several new interpretations.

Old data by Korablev ([51,52,53]) were reinterpreted, and vascular loops are proposed to be formed from one vessel but not from two vessels simultaneously.As a new explanation of sprouting, we assumed that two ECs simultaneously participate in the growth of sprouts, occurring in both ECs at once and not in one leading EC, as written in most literature reviews.A combined angiogenesis model is proposed. The model is composed of the formation of a loop and sprouts from the curved part of the loop and from one vessel and two participating ECs.A new interpretation of the splitting mechanism based on new features of vessel splitting proposed in the paper by Zaitseva et al. (2024; current issue) is proposed, namely, a new explanation for angiogenesis characterized by the splitting of blood vessels due to their compression; this is written in our article on angiogenesis in intestinal villi.A new interpretation of sprouting based on the participation of two ECs is proposed.It is proposed that, in situ (with the exception of nervous tissue), filopodia are not used; instead, invadopodia are formed.

However, many questions remain. Why do the leading ECs not divide, but only ECs from the STEM zone? What causes their division? Perhaps this is due to the fact that the leading ECs remove gap junctions from their contact zones, and this leads to the fact that the number of gap junctions in the EC cisternae, which are located immediately behind the leading ECs, decreases, which leads to a drop in the level of contact inhibition and commutes them to mitosis. Referring to the analogy of the division of ECs in the regenerating endothelium of the aorta, it is necessary to carry out 3DEM analysis and sprout markers.

There is a temptation to draw many diagrams containing various molecular machines characterized in vitro. Indeed, the vast majority of information concerning the function of the molecular machines involved in the process of angiogenesis has been obtained in vitro. The correspondence of their function to the processes taking place in situ has not been proven. We offer new models here that have not yet been fully tested in situ. Therefore, applying the information obtained about molecular machines that have not been tested in situ to new models that also need to be tested in situ would be the height of charlatanism. In this article, we have tried to draw attention to a strange situation where the phenotypic changes associated with the process of angiogenesis are practically completely unknown to the scientific community. For this, we would like to propose such an analogy: one cannot speculate about where and how various lubricating oils are located in engines without knowing how the engine itself works.

## Figures and Tables

**Figure 1 ijms-25-08575-f001:**
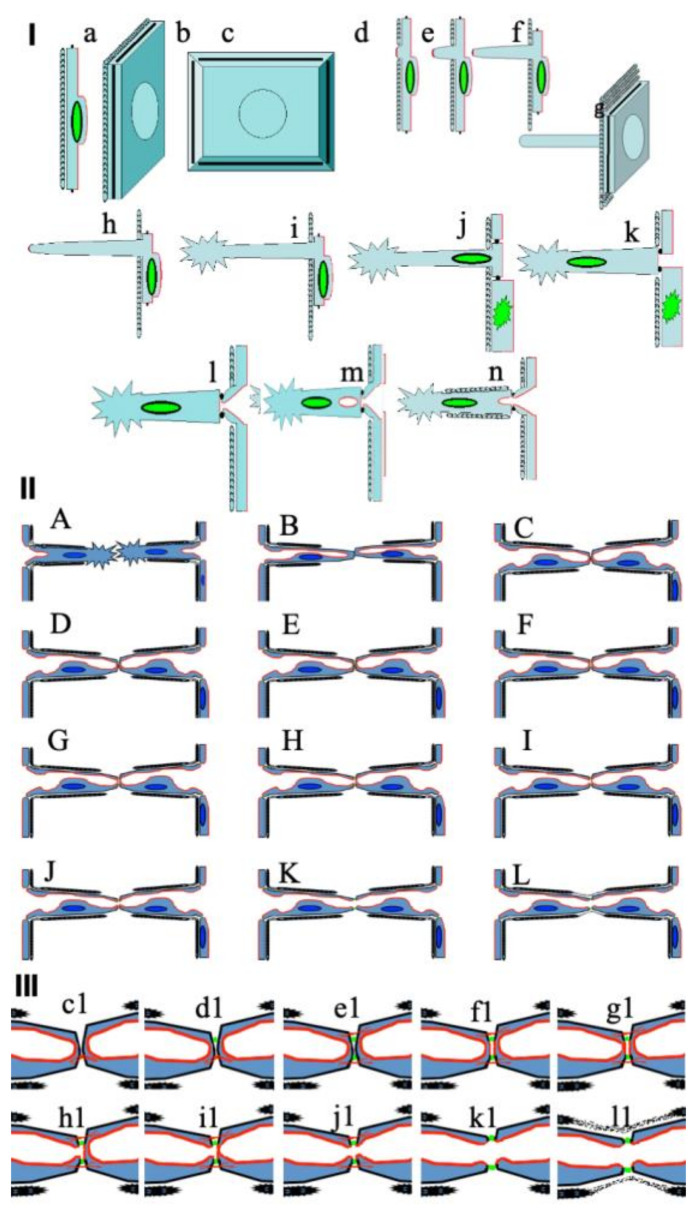
Scheme of consecutive steps of angiogenesis (adapted from Carmeliet et al. [18]). **I**. Stages of sprouting. **I** (**a**–**d**) The scheme of an EC. **I** (**e**–**g**) The formation of an invadopodium on the BLPM of the EC. This invadopodium dissolves the BM. **I** (**h**) The migration of this EC toward the invadopodium. Invadopodia are APM derivatives (or domains of the APM within the BLPM). **I** (**i**) The formation of a filopodium on the tip of the invadopodium. **I** (**j**) The shift of the EC nucleus inside the invadopodium. **I** (**k**) The whole EC is inside the sprout. The second stage of sprout elongation. **I** (**l**) The whole EC and parts of the neighboring ECs are inside the sprout. **I** (**m**,**n**) The formation of a new lumen and BM. The lumen of the capillary is formed by a portion of flat polarized ECs. The short stage of procollagen synthesis and transport for BM generation should take place. The formation of intracellular vacuoles from the APM dissolved in the BLPM induces the transport of procollagen and other extracellular matrix proteins and the formation of the BM. **II** (**A**–**L**) A scheme of the stages of canalization and the formation of the lumen within the contact between two sprouts. **III** (**c1**–**l1**) Higher magnification of the zone of contact between two sprouts. There are two possibilities: 1. This cell has lost the APM. 2. This cell retains a small APM. The cell then forms a tip with a filopodium, which explores the extracellular matrix, looking for channels in it. In the first case, the APM components are subjected to sequestration into apical vacuoles. In the second case, in the initial stage, there will be a generation of tight contacts and then the sequestration of the APM.

**Figure 2 ijms-25-08575-f002:**
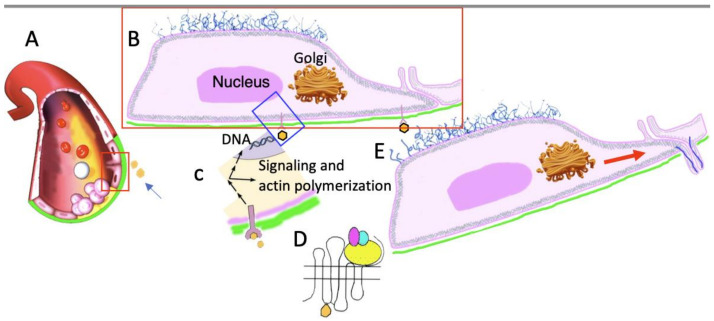
Rough scheme of cell signaling involved in angiogenesis in situ. (**A**) Ligands (arrow) or growth factors arrive at the basolateral surface of an EC in the blood vessel. The basement membrane is colored in green. (**B**) The EC and its receptors are shown (enlargement of the red box in (**A**)). The glycocalyx is visible on the apical plasmalemma. (**C**) Higher magnification of the box in (**B**). Interaction of the ligand with its receptor induces cellular events and signals to the nucleus, which would stimulate cell division. (**D**) The scheme of the interaction between a ligand and a receptor on the basolateral surface of the EC. Some adaptors (colored in magenta, yellow, and aqua) regulate the phosphorylation of proteins. (**E**) Two neighboring ECs form invadopodia. Membranes and metalloproteases (red arrow) for their formation and for the dissolution of the basement membrane are sent by the Golgi. Blue lines are microtubules. Invadopodia from two ECs form a sprout.

**Figure 3 ijms-25-08575-f003:**
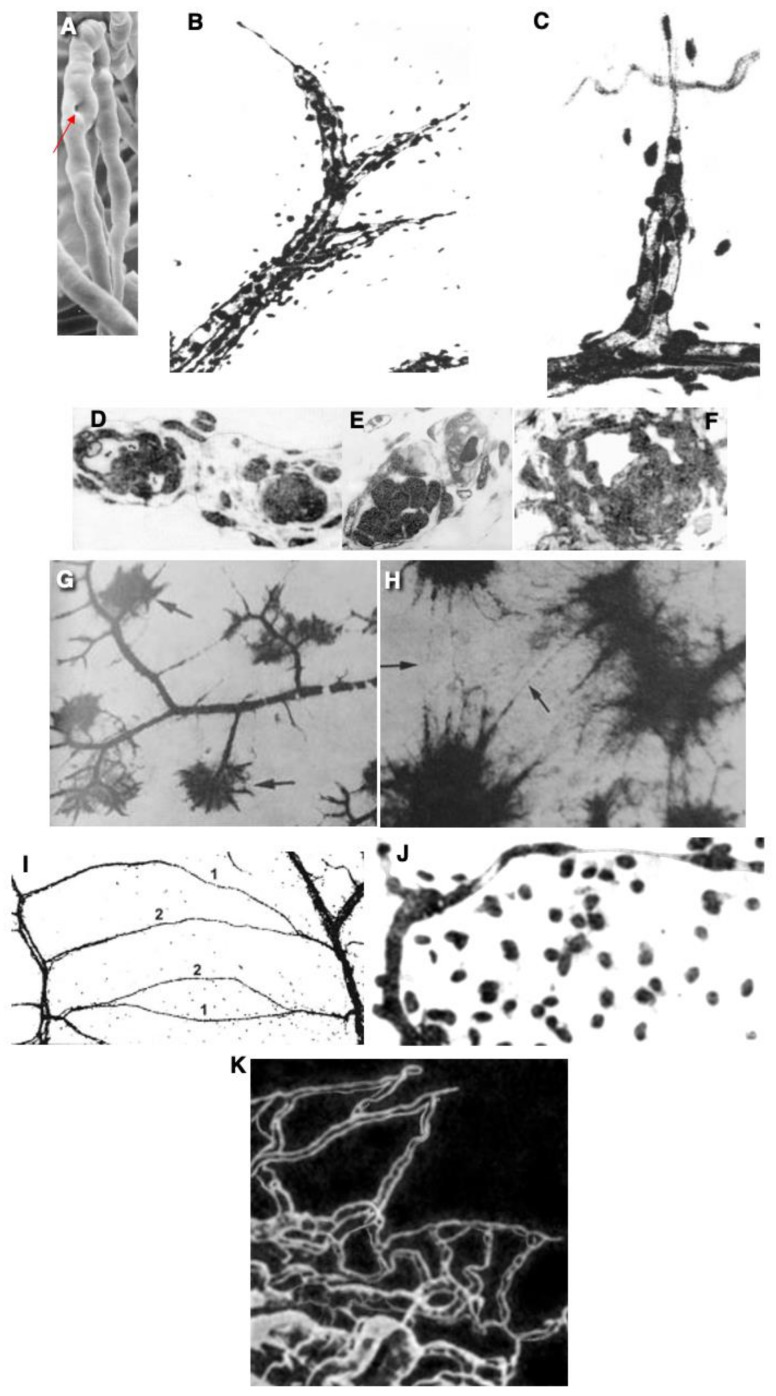
The structure of microvessels during ansiform angiogenesis. The large omentum of a human fetus, 22–26 weeks. (**A**) An example of intussusceptive angiogenesis. The hole (red arrow) in the plastic cast, which filled the lumen of a microvessel, indicates a pillar. (**B**) Three loop-like sprouts formed from the pre-existing one. (**C**) Higher magnification of a loop-like sprout. (**D**–**F**) Semi-thin sections of the vessel loop. (**D**) Sections (left and right) of two vessels near the base of the loop-like sprout. (**E**) A section of the loop-like sprout near its tip. (**F**) The cross-section of the loop-like sprout through its tip. Vessels in images (**B**,**C**,**G**–**J**) were subjected to impregnation with silver nitrate. (**G**,**H**) The centers of the loop-like sprouts are shown. (**H**) The beginning of the formation of a capillary network based on vascular loops. Higher magnification. (**I**,**J**) The coalescence of the sprouts. The formation of vascular bridges. (**K**) Scanning electron microscopy of a plastic vascular cast that filled the newly formed vessel in the cornea. The arrow shows the sprout growing from the tip of the vascular loop. Scanning electron microscopy of corrosion casts of the newly formed vessels growing in a cornea. There are several loop-like sprouts and splitting. Magnifications: ×400 (**A**,**C**); ×200 (**B**,**J**); ×900 (**D**–**F**); ×32 (**G**,**I**); ×100 (**H**). Image (**A**) is taken from [51]. Images (**B**,**D**–**F**) are from [52]. Images (**C**,**G**–**J**) are from [53]. Figures reprinted courtesy of Creative Commons License (Attribution–Non-commercial–Share Alike 4.0 Unported license).

**Figure 4 ijms-25-08575-f004:**
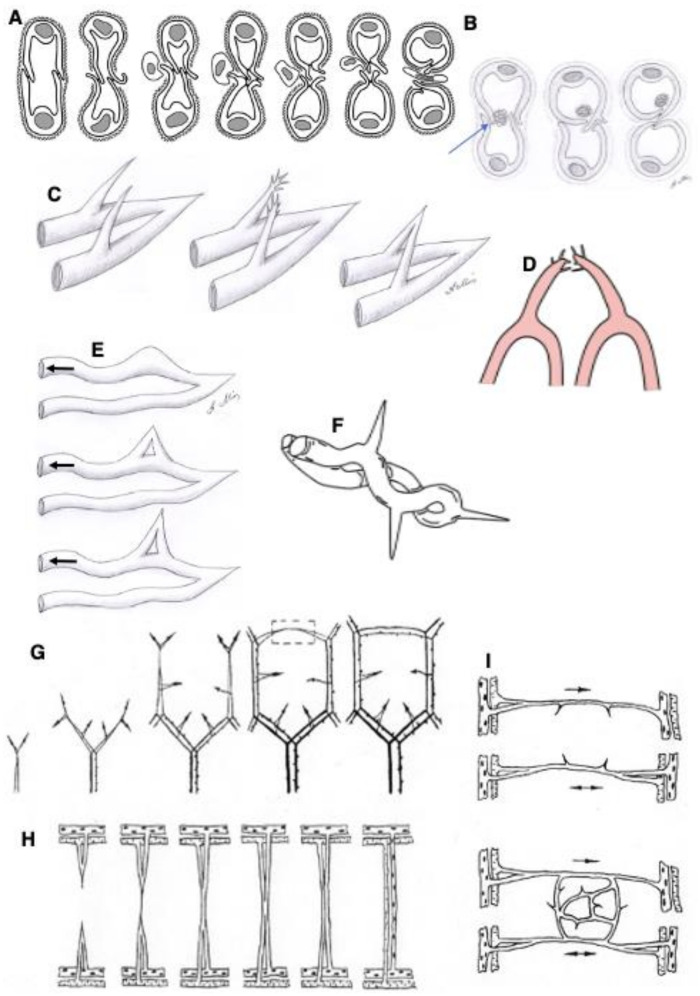
Schemes explaining possible mechanisms of intussusceptive angiogenesis (**A**) and angiogenesis based on the formation of vascular loops. (**B**) A scheme of intussusceptive angiogenesis based on the formation of a microthrombus (arrow). (**C**) The formation of a loop-like sprout based on the assumption that two independent sprouts are formed on both microvessels and generate sprouts and that these sprouts then coalesce and form a loop. (**D**) The formation of a loop from two pre-existing loops. (**E**) The formation of a loop-like sprout based on the inertial formation of a loop on the venous knee of the vascular loop. (**F**) The formation of loop-like sprouts toward different directions from the venous part of the pre-existing loop, which forms a spiral. (**G**–**I**) These images show how a vascular network could be formed after the fusion of loop-like sprouts.

**Figure 5 ijms-25-08575-f005:**
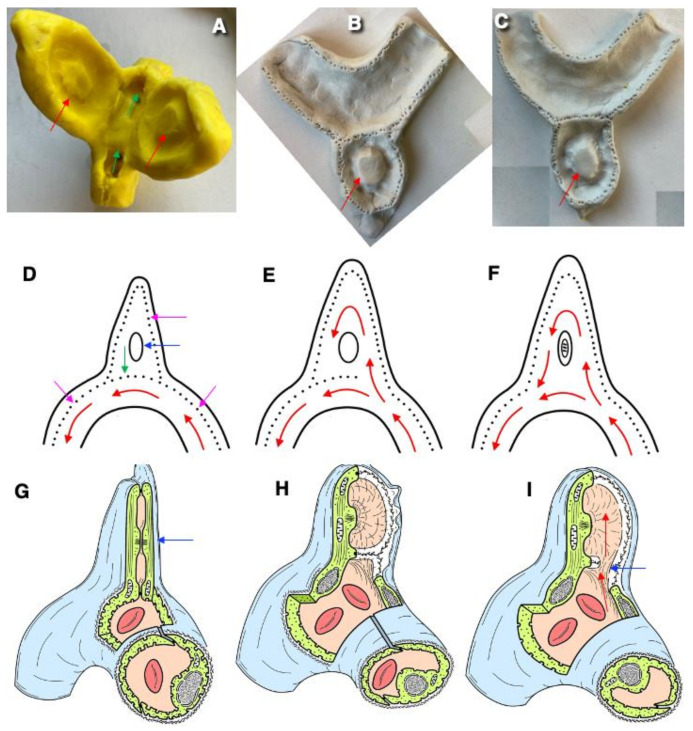
A scheme of the formation of the new vascular loop on the tip of a pre-existing vascular loop according to loop-dependent angiogenesis. (**A**–**F**) Plasticine models. (**A**) Hypothetical mechanisms of sprout formation from the loop of a microvessel before the formation of a new direction of blood flow. Green arrows show the direction of blood flow in the parent vessel. Red arrows demonstrate the island of the APM, where two ECs are tightly attached to each other. (**D**–**F**) A scheme of local blood flow. Magenta arrows show tight junctions. The green arrow indicates the tight junction separating the future channel from the existing blood flow. Red arrows show the direction of the blood flow in the parental vessel. In (**E**), the local destruction of tight junctions induced a new channel for the loop-like bloodstream. (**G**–**I**) A drawing of this model. (**G**) A cross-section of the sprout. The blue arrow indicates that island. (**H**) Another view of this sprout. (**I**) The blue arrow shows the site where the tight junction is disrupted. Red arrows indicate the new direction of blood flow.

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
