# Peer review of "New Interpretations for Sprouting, Intussusception, Ansiform, and Coalescent Types of Angiogenesis"

_ijms, 2024, doi:10.3390/ijms25168575_

Round 1

Reviewer 1 Report

Comments and Suggestions for Authors

The review entitled Angiogenesis: Old facts and new interpretations summarized the progress of angiogenesis. However, to my knowledge, I did not get any new findings from this review. All the figures were copied from the published data, and the authors did not generate any from their view. I suggest the authors bring more new research to the readers. Angiogenesis is important for many aspects, including cancer, wound healing, etc. It is valuable to figure out new molecular mechanisms of angiogenesis. I suggest the authors propose some new mechanisms of different aspects of angiogenesis, and draw some nice pictures accordingly.

Comments on the Quality of English Language

There are some typoes.

Author Response

Review 1

The review entitled “Angiogenesis: Old facts and new interpretations” summarized the progress of angiogenesis. However, to my knowledge, I did not get any new findings from this review. All the figures were copied from the published data, and the authors did not generate any from their view. I suggest the authors bring more new research to the readers. Angiogenesis is important for many aspects, including cancer, wound healing, etc. It is valuable to figure out new molecular mechanisms of angiogenesis. I suggest the authors propose some new mechanisms of different aspects of angiogenesis, and draw some nice pictures accordingly.

Our reply

  1. We are very glad that we came across a very competent and well-versed reviewer, who also excellently knows Russian scientific literature. However, unfortunately, two fundamental articles by A. Korablev published in 1988 and 1990 were practically inaccessible to the English-speaking reader at a time when machine translation from Russian into English was not developed. Only abstracts of these papers are available. The book "Angiogenesis" (22. Kupriyanov et al., 1993; see reference within the paper) is not translated into English and was not sent to Western libraries. These articles by Korablev even now were not accessible to the English-speaking scientists and also were firmly forgotten. At least, the Google scholar shows that this 1988 article, published in Russian, is not included in this database, and the same article, introduced as an English translation, has no citations. In addition, we did not find a citation of the article by Korablev (1988) in any literature review on angiogenesis available on PubMed. The existence of Korablev's hypothesis about such angiogenesis is not found in any of the literature reviews available on Pubmed.

In our opinion, the re-introduction of these results, is very important for scientists working in the field of angiogenesis. We think that the pioneering work in the field of angiogenesis should be known to readers.

  1. We agree with our reviewer that it is not completely clear what our interpretations are is really new. However, actually, in our paper there are several new interpretations.
  2. Old data by Korablev (1988; 1990) were reinterpreted and vascular loops are proposed to be formed from one vessel but not from two vessels simultaneously.
  3. A new explanation of sprouting: we assumed that two ECs simultaneously participate in the growth of sprouts occurs in two at once, and not in one leading EC, as written in most literature reviews. This idea is based on images presented by Wakui [65].
  4. The combined angiogenesis model is proposed. The model is composed of formation of a loop and sprouts from the curved part of the loop and from one vessels and two ECs participation.
  5. New interpretation of splitting mechanism based on new features of vessel splitting proposed in the paper by Zaitzeva et al (current issue) were proposed, namely, a new explanation has been proposed for angiogenesis by splitting blood vessels due to their compression, this is written in our article on angiogenesis in the intestinal villi.
  6. A new interpretation of sprouting based on participation of two ECs is proposed.
  7. It was proposed that in situ filopodia are not used; instead invadopodii are formed.

Also, we changed the title and added several new schemes.

Reviewer 2 Report

Comments and Suggestions for Authors

The work entitled “Angiogenesis: Old facts and new interpretations” by the authors Alexander V Korablev, Irina S Sesorova, Vitaly V. Sesorov, Pavel Sergeevich Vavilov, Anna Mironov, Anna V Zaytseva, Eugeny V Bedyaev and Alexander A Mironov is an interesting opinion article which deals with the different mechanisms of angiogenesis, relating old concepts with new facts and interpretations.  Numerous relevant data are considered. However, I have major considerations that need to be corrected.

 Frequent concepts are repeated throughout the work, using the same or similar words even in adjacent or near lines. For example, in lines 33 to 35, ("Currently, most of the molecular mechanisms involved in angiogenesis have been discovered". "Now most molecular machines involved into angiogenesis have been discovered".)  In lines 37 and 38 (" However, these are obtained mainly in cell culture and are not tied to real structures in tissues". " However, they are obtained mainly in cell culture and are not tied to real structures in tissues"). Therefore the text must be extensively revised.

Citation 1 is missing from References, starting with 2.

The citations in the text do not coincide with those in the references, probably due to final readjustments that were not reviewed again.

In the correlation between old facts and new interpretations, "Ansiform (by loop formation) angiogenesis" is striking. In this concept, the works of Patan and other later authors that relate loop formation with intussusception should be taken into account.

In conclusion, it seems that the article has been rewritten without a definitive correction. If a thorough review of the manuscript were now undertaken, this reviewer would be interested in re-evaluating the work.

Author Response

Reply Reviewer 2

The work entitled “Angiogenesis: Old facts and new interpretations” by the authors Alexander V Korablev, Irina S Sesorova, Vitaly V. Sesorov, Pavel Sergeevich Vavilov, Anna Mironov, Anna V Zaytseva, Eugeny V Bedyaev and Alexander A Mironov is an interesting opinion article which deals with the different mechanisms of angiogenesis, relating old concepts with new facts and interpretations.  Numerous relevant data are considered. However, I have major considerations that need to be corrected.

  1. Frequent concepts are repeated throughout the work, using the same or similar words even in adjacent or near lines. For example, in lines 33 to 35, ("Currently, most of the molecular mechanisms involved in angiogenesis have been discovered". "Now most molecular machines involved into angiogenesis have been discovered".)  In lines 37 and 38 (" However, these are obtained mainly in cell culture and are not tied to real structures in tissues". " However, they are obtained mainly in cell culture and are not tied to real structures in tissues"). Therefore the text must be extensively revised.

Our reply

Thanks a lot for your work. We eliminated repetitions

  1. Citation 1 is missing from References, starting with 2.

Our reply

We corrected this mistake.

  1. The citations in the text do not coincide with those in the references, probably due to final readjustments that were not reviewed again.

Our reply

We corrected this mistake.

  1. In the correlation between old facts and new interpretations, "Ansiform (by loop formation) angiogenesis" is striking. In this concept, the works of Patan and other later authors that relate loop formation with intussusception should be taken into account.

Our reply

We quoted Patan et al. However, our attempts to find in the later reviews of this type of angiogenesis with Patan et al quoting were unsuccessful. Patan et al., (1996) concluded that during the growth of the human colon adenocarcinoma in vivo, intussusception is an important mechanism of tumor angiogenesis. In 2000, Patan (2000; doi: 10.1023/a:1006493130855) stated that angiogenesis comprises two different mechanisms: endothelial sprouting and intussusceptive microvascular growth. He did not mention the ansiform growth of blood vessels described by Korablev (1988; PMID: 2453188). After three-dimensional reconstruction of cancer using histological serial sections sections, Patan et al. (2001; doi: 10.1006/mvre.1996.0025) revealed a new vascular network composed of venous-venous loops of varying sizes grows inside the tumor from the wall of the adjacent main vein. The intussusceptive microvascular growth is responsible for this. However, this mechanism is rather different from the ansiform growth of blood vessels. In his consecutive paper, Patan (2004; doi: 10.1007/978-1-4419-8871-3_1) states that angiogenesis comprises two different mechanisms: endothelial sprouting and intussusceptive microvascular growth. In this paper, the formation of loops by newly formed venous vessels was not mentioned. This loop-based model is not mentioned by Dubley (2012; doi: 10.1101/cshperspect.a006536.) and by Zhang et al. (2024; doi: 10.3389/fonc.2024.1359069). Even in the Russian papers, the ansiform growth of blood vessels was not mentioned (Tashkov et al., 2015, https://doi.org/10.24884/1682-6655-2015-14-4-11-17; Sheiko et al., 2022; https://doi.org/10.17650/2782-3687-2022-14-2-28-35).

  1. In conclusion, it seems that the article has been rewritten without a definitive correction. If a thorough review of the manuscript were now undertaken, this reviewer would be interested in re-evaluating the work.

Our reply

We have improved our paper.

Round 2

Reviewer 1 Report

Comments and Suggestions for Authors

This revision sounds better than the first version with good improvement on the recent research. The conclusion section points out the future directions on angiogenesis. However, without good presentation like figures, the readers can not get the recent progress of angiogenesis. I strongly suggest the authors to add figures on the signaling pathways rather than the figures about phenotype.

Comments on the Quality of English Language

NA.

Author Response

Reviewer 1:

This revision sounds better than the first version with good improvement on the recent research. The conclusion section points out the future directions on angiogenesis. However, without good presentation-like figures, the readers cannot get the recent progress of angiogenesis. I strongly suggest the authors to add figures on the signalling pathways rather than the figures about phenotype.

Our reply

The progress in the analysis of molecular mechanisms involved into angiogenesis is out of our focus. We specifically stress this circumstance in Introduction and in Conclusion. Moreover, our paper is not a review but just an opinion. On the other hand, we understand that there is a temptation to draw many diagrams where to place various molecular machines obtained in vitro. Indeed, the vast majority of information concerning the function of the molecular machines involved in the process of angiogenesis has been obtained in vitro. The correspondence of their function to the processes taking place in situ has not been proven. We are offering new models here that have not yet been fully tested in situ. Therefore, stringing the information obtained about molecular machines that have not been tested in situ onto new models that also need to be tested in situ will be the height of charlatanism. This situation is well known in the field of intracellular transport. For example, although it is believed that all cells have exit sites from the ER, we did not find them in enterocytes and hepatocytes. In this article, we tried to draw attention to the strange situation when the phenotypic changes associated with the process of angiogenesis are practically completely unknown to the scientific community. For this we would like to propose such an analogy. One can't fantasize about where and how various lubricating oils are located in engines without knowing how the engine itself works. Nevertheless, we tried to satisfy the reviewer and, at least with minimal involvement of our imagination, propose an additional scheme of angiogenesis (Now: Fig. 2) indicating some steps of cell signalling.

Reviewer 2 Report

Comments and Suggestions for Authors

The work has been improved.

 In the latest version, the sentences included in lines 35 and 36 could be incorporated into a single one. For example "Currently, most of the molecular mechanisms and machines have been discovered."

There are minor typographical errors that can be corrected by the publisher. For example: double spacing or some words (e.g. spouts instead of sprouts -line 603).

References have been added and others have been removed. We consider that those deleted should be incorporated again into the final manuscript.

Author Response

Reviewer 2: The work has been improved.

1. In the latest version, the sentences included in lines 35 and 36 could be incorporated into a single one. For example, "Currently, most of the molecular mechanisms and machines have been discovered."

Our reply

We corrected this according to this proposal.

2. There are minor typographical errors that can be corrected by the publisher. For example: double spacing or some words (e.g. spouts instead of sprouts - line 603).

Our reply

We corrected this according to this proposal.

  1. References have been added and others have been removed. We consider that those deleted should be incorporated again into the final manuscript.

Our reply

We corrected this according to this proposal. However, due to significant changes in the text, the full realization of this demand is almost impossible. We included these references in the end (references [72-75]).

Round 3

Reviewer 1 Report

Comments and Suggestions for Authors

No further comments.